# Isoniazid and Rifampicin Resistance-Conferring Mutations in *Mycobacterium tuberculosis* Isolates from South Africa

**DOI:** 10.3390/pathogens12081015

**Published:** 2023-08-04

**Authors:** Afsatou Ndama Traoré, Mpumelelo Casper Rikhotso, Marry Avheani Mphaphuli, Sana Mustakahmed Patel, Hafsa Ali Mahamud, Leonard Owino Kachienga, Jean-Pierre Kabue, Natasha Potgieter

**Affiliations:** Department of Biochemistry and Microbiology, Faculty of Sciences, Engineering & Agriculture, University of Venda, Thohoyandou 0950, South Africa; mputso@yahoo.com (M.C.R.); marryavheani78@gmail.com (M.A.M.); patelsana9803@gmail.com (S.M.P.); hafsa.mahamud98@gmail.com (H.A.M.); leokachienga@gmail.com (L.O.K.); kabue.ngandu@univen.ac.za (J.-P.K.); natasha.potgieter@univen.ac.za (N.P.)

**Keywords:** tuberculosis, INH, RIF, resistance-conferring mutations, drug resistance mechanisms

## Abstract

Tuberculosis (TB), caused by *Mycobacterium tuberculosis* (*M. tb*), remains a significant global health issue, with high morbidity and mortality rates. The emergence of drug-resistant strains, particularly multidrug-resistant TB (MDR-TB), poses difficult challenges to TB control efforts. This comprehensive review and meta-analysis investigated the prevalence of and molecular insights into isoniazid (INH) and rifampicin (RIF) resistance-conferring mutations in *M. tb* isolates from South Africa. Through systematic search and analysis of 11 relevant studies, we determined the prevalence of gene mutations associated with RIF and INH resistance, such as rpoB, katG, and inhA. The findings demonstrated a high prevalence of specific mutations, including S450L in rpoB, and S315T, which are linked to resistance against RIF and INH, respectively. These results contribute to the understanding of drug resistance mechanisms and provide valuable insights for the development of targeted interventions against drug-resistant TB.

## 1. Introduction

Tuberculosis (TB), caused by *Mycobacterium tuberculosis* (*M. tb*), continues to be a major global health issue [1]. It is the highest cause of mortality caused by a single infectious agent, surpassing human immunodeficiency virus/acquired immunodeficiency syndrome HIV/AIDS [2]. Despite global reductions in TB incidence and death during the last few decades, in 2019, it was estimated that 10 million people were infected with TB and 1.41 million people died from the disease [1].

In many low- and middle-income countries, TB/HIV co-infection and the emergence of MDR-TB strains have become significant obstacles in the fight against TB [3]. Multidrug-resistant TB (MDR TB) is caused by TB bacteria that are resistant to at least isoniazid and rifampin [1]. Antimycobacterial drug resistance is threatening TB prevention and control efforts, and TB continues to be a major public health problem on a global scale [3]. Rifampicin (RIF), also known as rifampin, is an ansamycin antibiotic used to treat several types of bacterial infections, including tuberculosis. Isoniazid (INH), also known as isonicotinic acid hydrazide, is an antibiotic used for the treatment of tuberculosis. The *rpoB* gene codes for the RNA polymerase β subunit, which is the target of rifampicin. *KatG* is an enzyme that functions as both catalase and peroxidase. Its mutation is the cause for Mycobacterium (specifically *M. tuberculosis*) resistance to the drug isoniazid, which targets the mycolic acids within the tuberculosis bacteria. InhA (enoyl-ACP reductase) is an essential component of the mycobacterial FAS-II system responsible for mycolic acid synthesis, particularly in *M. tuberculosis*. In 2019, there were approximately 500,000 new cases of RIF-resistant TB globally, 78% of which were MDR-TB [1]. *M. tb* drug resistance develops because of spontaneous gene mutations that limit the bacterium’s sensitivity to the most-used anti-TB drugs. These genes can encode drug targets or drug metabolism pathways, which can affect the efficacy of anti-TB therapy [1]. 

Numerous previous studies [4,5,6] identified various genes that encode anti-TB drug targets and briefly discussed various mechanisms of resistance to RIF and INH. More than 95% of RIF resistance is associated with *rpoB* gene alterations in an 81-bp area. INH resistance appears to be more complex and has been linked to numerous genes, most notably *katG* and the *inhA* promoter region. The most frequent gene mutations causing RIF and INH resistance in *M. tb* have not yet been evaluated in a comprehensive study and meta-analysis in South Africa. To further understand the overall proportion of phenotypic INH and RIF resistance explained by existing single or canonical gene mutations, the estimated pooled prevalence of RIF resistance-associated gene mutations, as well as the frequencies of co-occurring or multiple mutations, were investigated in the current review. 

Understanding the incidence and prevalence of drug resistance-conferring mutations linked to RIF- and INH-resistant *M. tb* is thus crucial. To determine the incidence and predominance of the most prevalent gene mutations linked to phenotypic RIF and INH resistance in *M. tb*, a comprehensive review and meta-analysis was conducted in South Africa.

## 2. Methods 

### 2.1. Study Protocol

The research was carried out using the Preferred Reporting Items for Systematic Reviews and Meta-analyses (PRISMA) protocol to search records from online databases, perform paper screening by title and abstract, and conduct assessment of the full-text suitability for systematic review and meta-analysis (Figure 1).

### 2.2. Databases and Search Strategy 

PubMed, MEDLINE, Web of Science, Scopus, Cochrane Library, and Google Scholar electronic sources were searched for articles written in English, for a period of five years (2018–2022). Studies that reported gene mutations conferring RIF and INH resistance in *M. tb* in South Africa were included in the analysis. The following specific key terms for databases searching were used: *Mycobacterium tuberculosis* OR tuberculosis AND INH OR isoniazid AND RIF OR rifampicin AND resistance OR resistant AND mutations OR sequence AND South Africa. A total of 5243 studies were retrieved from the database, as shown in Table 1. Overall, 11 studies were selected for inclusion in the review based on the inclusion criteria, whereas a total of 5232 were eliminated from the review due to not being eligible for inclusion.

### 2.3. Inclusion and Exclusion Criteria

The review conducted a comprehensive analysis of observational studies in South Africa that focused on diagnosing resistance to Rifampicin (RIF) and Isoniazid (INH) in *Mycobacterium tuberculosis* (*M. tb*) using approved molecular drug susceptibility testing (DST) tools recommended by the World Health Organization (WHO). Our study aimed to identify gene mutations associated with RIF and INH resistance and to determine the prevalence of drug resistance in different forms of tuberculosis (TB) cases.

The review included the studies that met the following inclusion criteria: studies reporting mechanisms of anti-TB drug resistance or gene mutations associated with RIF and INH resistance in *M. tb*, providing data on the prevalence of anti-TB drug resistance among TB patients, including both newly diagnosed and re-treated cases, utilizing standard WHO-approved molecular DST tools for TB diagnosis, and published research conducted in South Africa and available in the English language. The review excluded studies that did not meet the following exclusion criteria: studies that did not report mechanisms of anti-TB drug resistance or gene mutations associated with RIF and INH resistance, studies that did not focus on TB drug resistance prevalence in TB patients, studies that did not use standard WHO-approved molecular DST tools, and studies conducted outside South Africa or published in languages other than English.

### 2.4. Data Extraction 

The following data were extracted from the inclusion studies: author(s) name; year of publication; study period; study region; type of TB patients; study design; molecular DST method(s); sample size; total positive cases; total *M. tb* isolates for which DST was performed; frequency of any anti-TB drug resistance, any INH or RIF resistance, and MDR-TB; and gene mutations associated with RIF and INH resistance.

### 2.5. Meta-Analysis 

The meta-analysis was undertaken with the primary objective of providing transparent, objective, and reproducible summaries of the study outcomes (Figure 2). In the statistical analysis of data obtained from cohort, medical, and intervention studies, odds ratios were employed as the measure of choice to assess the strength of the association between events and their respective outcomes. Through this meta-analysis, we investigated the association between the prevalence of positive cases and resistance to Rifampicin (RIF) and Isoniazid (NIH), in conjunction with gene mutations (Figure 2).

The magnitude of the association between the occurrences was assessed using odds ratios (ORs). A positive link between the variables was established when the OR exceeded zero; conversely, if the OR was less than zero, no positive association was observed. The findings of the meta-analysis, as depicted in Figure 2, revealed a notable level of heterogeneity among the included studies. The study also found heterogeneity which indicated that there were substantial variations in the results and methodologies among the studies incorporated in the meta-analysis (Figure 2). This heterogeneity could have been the differences in study populations, data collection methods, or study designs.

R Programming was used to carry out statistical analysis. Random effects models were used to estimate the overall effect size for variation between studies and to provide estimates of heterogeneity. The “rma” function from the package “Metafor” was used for conducting meta-analyses and estimating the overall effect size. Heterogeneity was measured, and Cochran’s Q test was used to assess heterogeneity in the meta-analysis. A forest plot of the meta-analysis was generated to display the effect sizes and confidence intervals of inclusion studies. The inclusion studies variability was determined statistically in the R statistical tool. The heterogeneity measure (I^2^) provided an estimate of the proportion of variability in a meta-analysis that is explained by differences between the included studies rather than by sampling error.

## 3. Results 

### 3.1. Search Results 

As shown in Figure 1, a total of 5243 studies were retrieved through the search engines. Of the overall studies, a total of 5232 were eliminated from the review due to not meeting the criteria. Only 11 studies on the prevalence of gene mutations associated with RIF- and INH-resistant *M. tb* in South Africa were included in the review.

### 3.2. Characteristics of Studies Included

The studies, as shown in Table 1, were eleven in total, from which eight were conducted in Western Cape, two were from Free State, KwaZulu-Natal and the remaining studies were from Gauteng, Mpumalanga, Northwest and Limpopo. Most of these studies (7/11; 63.6%) were conducted between 1–3 years. The majority of the studies (10/11; 90.9%) were of pulmonary TB patients and 2/11 (18.2%) studies had retrospective studies, 2/11 (18.2%) prospective studies, 1/11 (9.1%) a retrospective cohort study, 1/11 (9.1%) a prospective cohort study, 1/11 (9.1%) a prospective multi-center diagnostic study, 1/11 (9.1%) a multi-center study, 1/11 (9.1%) a descriptive study, and 1/11 (9.1%) a multicenter observational study. Whole-genome sequencing method (Illumina HiSeq 2500) was used in 4/11 (36.4%) studies. A total of 3/11 (27.3%) studies used GeneXpert MTB/RIF, GenoType MTBDRplus, and BD MAX MDR-TB assay, making them the most common molecular DST methods used.

According to Table 1, the total number of patients who participated was 51,623. Among those who participated, 43,580 (84%) patients were found to be M-TB positive, and the number of isolates obtained after performing DST was 650 (1.2%). Among the positive cases, a total of 3637 (7%) cases were found to be drug resistant TB. In addition, a total of 2995 (6%) of the cases were INH-resistant, 3460 (7%) were RIF-resistant, and 2909 (6%) were MDR. The mutations for *katG* were found to be 885 (2%), and 2457 (5%) patients had mutations in the inhA promoter region. Furthermore, *katG* + *inhA* mutations were detected in 404 (1%) patients, while none of the samples had mutations on both *rpoB* + *katG* genes [Table 1].

### 3.3. Prevalence of any Rifampicin (RIF) or Isoniazid (INH) Resistance in Mycobacterium tuberculosis Isolates

The variance of the anti-TB resistance differs from geographical location and economic status of the populations of the country. The prevalence of any anti-TB drug resistance varied among the studies and geographical locations in South Africa. The study also revealed that 4/11 (36.4%) of the studies reported a prevalence of any anti-TB drug resistance ranging from 0.5 to 3%, with three of the studies reported in the Western Cape and two other studies reported in KZN and Free state, respectively, with a prevalence of less than 3%. A higher prevalence of over 80% was reported in studies from Gauteng, Mpumalanga, Northwest, and Limpopo (Table 1). In addition, the study further reported that the prevalence of any RIF and INH-resistant *M. tb* was below 15% in 10/11 (91%) studies from Western Cape, Free State, and KZN, with 1/11 (9%) reporting over 60% in studies from Gauteng, Mpumalanga, Northwest, and Limpopo [Table 1]. On the prevalence of MDR-TB, 2/11 (18%) studies reported over 30% from Western Cape and Free-state, respectively, while the remaining studies reported less than 10% prevalence of MDR-TB (Table 1).

### 3.4. Frequency of rpoB, katG and inhA promoter Mutations

A total of 2995 (6%) *M. tb* strains with any INH resistance were identified using standard WHO-approved molecular diagnostic methods, among which a higher proportion of mutations was detected in the *katG* gene 885 (2%) compared with the *inhA* promoter region 2457 (5%) according to Table 1. In addition, for RIF-resistant *M. tb* strains, the most common mutations were found in the following order: *rpoB* S450L probe (355 cases), Ile 491 Phe (35 cases), and L430P (34 cases), with the remaining probes reporting single cases each. 

## 4. Meta-Analysis

A meta-analysis was conducted to ensure transparent, objective, and replicable summaries of the study findings. The eleven included studies provided adequate information for statistical analysis. The results are shown in Figure 2, illustrating the variability in the prevalence of positive cases and resistance to RIF and NIH, as well as gene mutations, within the population investigated. The meta-analysis used the Q test (Cochran’s Q = 263.3585 [df = 10], *p* < 0.0001), revealing a significant outcome. Furthermore, the overall heterogeneity test yielded a heterogeneity measure (I^2^) value of 99.84% (*p <* 0.0001), indicating a highly significant level of heterogeneity in the findings.

As shown in Figure 2, odds ratios were measured for the included studies to investigate the association between the prevalence of positive cases and resistance to Rifampicin (RIF) and Isoniazid (NIH), along with gene mutations, within the population under investigation. The analysis of TB patients in relation to resistance and mutations demonstrated a significant association within the studied population.

The analysis of aggregated data in the study reveals strong and statistically significant positive associations, as shown in Figure 2. These findings are valuable as they enhance the current scientific knowledge in this specific area. The study results have important implications for future research, clinical practice, and the development of interventions related to the studied outcomes. Specifically, the findings suggest a significant relationship between the prevalence of positive cases and the presence of resistance to RIF and NIH, along with gene mutations, among TB patients in the examined population. Understanding these interrelationships is crucial for guiding further research and informing strategies for managing and treating tuberculosis effectively.

The level of heterogeneity among the studies included in the meta-analysis suggests significant variations in results and methodologies. These differences may arise from variations in study populations, data collection methods, or study designs. Consequently, interpreting the overall results requires caution, and further investigation into the sources of heterogeneity is warranted to enhance the robustness and applicability of the findings.

## 5. Discussion

The emergence of drug-resistant bacilli poses a significant challenge to global TB control and prevention efforts. The utilization of molecular-based diagnostic methods, which involve the detection of mutations in specific genes associated with anti-TB drug resistance, is recognized as a more efficient and effective approach.

The detection of gene mutations in resistance-determining regions within resistant *M. tb* isolates plays a crucial role in the rapid identification of anti-TB drug resistance. This approach not only assists in the timely detection of resistance but also facilitates the exploration of resistance mechanisms, thereby aiding in the development of effective strategies to combat drug-resistant tuberculosis. Understanding the molecular mechanisms underlying drug resistance in *Mycobacterium tuberculosis* (*M. tb*) is crucial for the development of improved diagnostic tools. Further investigation is warranted to identify specific gene mutations that contribute to drug-resistant *M. tb*, particularly multidrug-resistant tuberculosis (MDR-TB). Such research will provide valuable insights for local tuberculosis (TB) control efforts and inform the development of effective strategies to combat MDR-TB within the country [16].

The meta-analysis aimed to provide transparent, objective, and reproducible summaries of study outcomes related to the association between the prevalence of positive tuberculosis (TB) cases and resistance to Rifampicin (RIF) and Isoniazid (NIH), along with gene mutations. Odds ratios were used as the measure to assess the strength of this association. The study revealed significant heterogeneity among the included studies, indicating variations in results and methodologies, possibly due to differences in study populations, data collection methods, or study designs.

The analysis demonstrated a notable positive association between TB cases and resistance to RIF and NIH, along with gene mutations, within the studied population. The findings have important implications for future research and clinical practice in managing and treating tuberculosis effectively. However, caution is advised in interpreting the overall results due to the observed heterogeneity, and further investigation into its sources is needed to enhance the reliability and applicability of the findings as a limitation of the review.

In this review, we assessed the prevalence of mutations in genes associated with RIF- and INH-resistant *M. tb* in South Africa. Our review demonstrated a prevalence of 885 (2%) *katG*, 2457 (5%) *inhA*, and 404 (1%) *katG* + *inhA* mutations in patients with TB in South Africa. The majority of the mutations were due to *rpoB*: L430P 34 (8%), S450L 355 (80%), and Ile 491 Phe 35 (8%). Other mutations were in the *katG* S315T 502 (99%) gene. The mutations occurred at different positions within the *rpoB* gene and resulted in alterations to the structure or function of the RNA polymerase enzyme. These changes interfere with the binding of RIF to the enzyme, rendering the drug less effective in inhibiting bacterial growth. In the *katG* gene mutations, S315T lead to reduced activation of INH, reducing the drug’s effectiveness in killing *Mycobacterium tuberculosis*.

In accordance with our findings, a previous systematic review conducted in Ethiopia corroborated our observations, revealing that S315T mutations in the *katG* gene accounted for 79.1% of INH resistance in *M. tb* isolates [17]. Consistent with this, a study conducted in India, which bears the highest burden of TB and multidrug-resistant TB (MDR-TB) globally, reported that among the tested isolates, 71.0% exhibited detectable mutations in the *katG* 315 region, while 29.0% exhibited mutations in the inhA promoter region [18]. A similar study conducted in Ethiopia reported a pooled prevalence of 63.2% for the *katG* MUT1 (S315T1) mutation [18]. Unfortunately, very little evidence from African studies supports these findings. A study conducted in Uganda demonstrated that *katG* and *inhA* gene mutations were primarily attributed to S315T (76%) and C15T (8%) nucleotide changes, respectively [19]. A recent study conducted in Monrovia, Liberia, indicated suggestively higher estimated global frequencies of *katG* 315 and inhA-15 at 86% and 34%, respectively [20]. It is estimated that approximately 64% of phenotypic resistance to INH globally can be attributed to the *katG* (S315T) mutation [21]. In the case of RIF resistant isolates, our study revealed that the most prevalent gene mutation associated with RIF resistance was observed at S450L 355 (80%). A study conducted in 2021 also reported that mutations in the *rpoB* S450L region were associated with high levels of resistance [22].

The origin of development of drug-resistant tuberculosis (DR-TB) and the transmission, whether direct or by other means, especially in developing countries, are triggered by factors such as immune-compromised HIV individuals, poor living conditions, poor sanitation and acceptable hygiene, poor healthcare administration (diagnostic tools and delaying drug susceptibility testing (DST) practices), inadequate administration of anti-TB therapy regimens and patient compliance, high prevalence of diabetes mellitus, alcoholism, and smoking [23,24,25]. Seid et al. [26] reported that the choice of GenoType MTBDRplus, MTBDRsl LPAs, and WGS as diagnostic techniques have proven to be efficient in the detection of specific genes that are responsible for triggering mutation by anti-TB drug resistance by reducing the timeframe in a matter of hours, instead of weeks or months as is traditionally known across the globe. The prevalence of anti-TB drug resistance among all diagnosed TB patients was 3637 (7%), while the prevalence of any RIF and INH resistance was 3460 (7%) and 2995 (6%), respectively [Table 1] in the study. Eddabra and Neffa [24] further stated that RIF resistance is prone to the mutations on the *rpoB* gene, especially in an 81 base pair region, and INH resistance frequently occurs on the *katG* gene. The variance of anti-TB resistance differs depending on geographical location and economic status of the populations of the country. This was further confirmed by the studies from the provinces of Western Cape and KZN of South Africa. In the study, a total of 2995 (6%) 1512 Mtb strains with any INH resistance were identified using standard WHO-approved molecular diagnostic methods, among which a higher proportion of mutations was detected in the *katG* gene 885 (2%), compared with the inhA promoter region 2457 (5%), according to Table 1. In RIF-resistant *M. tb* strains, the most common mutations were found in the following order: *rpoB* S450L probe (355 cases), Ile 491 Phe (35), and L430P (34 cases), with the remaining probes reporting single cases each. In a review by Seid et al. [26], it was reported that most of the mutations were found in codons 531 (34.01%), 526 (9.3%), and 516 (2.33%) in the RIF resistance-determining region (RRDR) of the *rpoB* gene, and those related to RIF-resistance were in codon 531, followed by 526, mutations. These findings were not far from the findings of this study as the codon’s points were not far apart. This slight variation is due to geographical location, health status of the populace, and the general economic status of the populations of the country. Biologically, this genetic variation occurs when there is a single base substitution with majority of common mutation in the *rpoB* gene encoding *β*-subunit of DNA-dependent RNA polymerase at codon 531 (S531L) [24]. In studies by Isakova et al. [27], Minh et al. [28], and Adikaram et al. [29] in Vietnam and Sri Lanka, respectively, over 95% and 30% exhibited mutations in the rpoB gene at codon 531 by RIF-resistant strains, respectively. In this study a higher proportion of mutations was detected in the *katG* gene, with 885 (2%), compared with the inhA promoter region, with 2457 (5%), as seen in Table 1. However, Seifert et al. [21] reported that there were over 300 (64%) mutations occurring in the *katG* and *inhA* genes in a promoter region of *inhA*. Seifert et al. [21] further opined that a majority of 315 mutations (64%) in the *katG* gene and 15 mutations in the inhA gene is the dominant (19%) mutation at the inhA promoter region, which was higher in relation to this study, where mutation on the same region was less than 2%. This frequency of mutation trends was at the *rpoB*, *katG*, and *inhA* genes in MDR-TB, which was like other studies carried out globally in countries such as Vietnam, Ethiopia, and Sri Lanka, among others [26,28].

## 6. Conclusions

The emergence of MDR-TB presents a significant global challenge to TB control and prevention efforts. To address this, molecular-based diagnostic methods that detect specific gene mutations associated with anti-TB drug resistance have been recognized as more efficient and effective approaches. The review was conducted to assess the prevalence of gene mutations related to resistance against RIF and INH in *M. tb* isolates. The study found that the most prevalent mutations in the *rpoB* gene were S450L (80%), Ile 491 Phe (8%), and L430P (8%), which result in structural or functional alterations in the RNA polymerase enzyme. These changes interfere with the binding of RIF to the enzyme, reducing its effectiveness in inhibiting bacterial growth. In the *katG* gene, the predominant mutations were S315T (99%), leading to decreased activation of INH and compromising its efficacy in killing *M. tb*. These findings are consistent with previous studies conducted in Ethiopia, India, Uganda, and Liberia, which also reported high prevalence of the S315T mutation in the *katG* gene and mutations in the *rpoB* gene associated with rifampicin resistance. This review contributed to the understanding of drug resistance mechanisms and provided valuable insights for the development of targeted interventions against drug-resistant TB.

Understanding the molecular mechanisms of drug resistance in *Mycobacterium tuberculosis* is of paramount importance for further developing improved diagnostic tools and effective strategies to combat MDR-TB. Further research is necessary to identify specific gene mutations that contribute to MDR-TB and to inform the development of tailored interventions.

## Figures and Tables

**Figure 1 pathogens-12-01015-f001:**
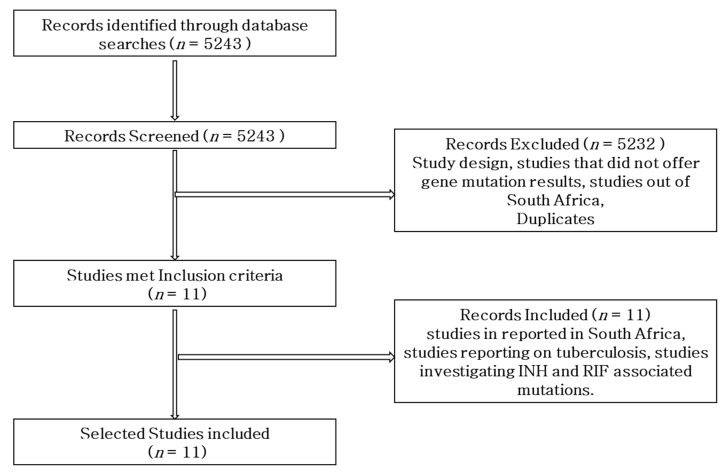
Flow diagram of the literature search strategy, search results, and inclusion and exclusion of articles.

**Figure 2 pathogens-12-01015-f002:**
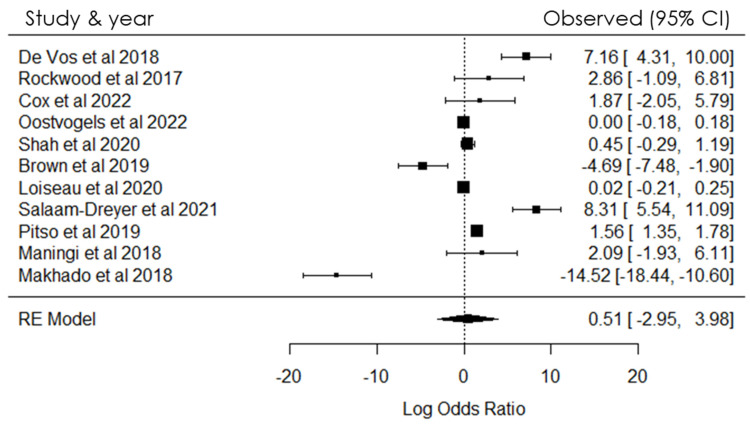
Forest plot showing heterogeneity among studies investigating gene mutations in SA [4,5,7,8,9,10,11,12,13,14,15].

**Table 1 pathogens-12-01015-t001:** Characteristics of studies included in the systematic review.

Author(s)	Study Region	Study Period	Type of patients (EXTRA or PTB)	Study design	Molecular Diagnostic Method(s)	Participants (*n*)	All Positive Cases (*n*)	Total Isolates with DST Performed (*n*)	Any Drug Resistance (*n*)	Any INH or RIF Resistance (*n*)		MDR-TB (*n*)	Anti-TB Drug Resistance or Mechanisms rpoB and KatG (*n*)	Frequency of Gene Mutations (*n*)			
										INH	RIF			katG	inhA	katG + inhA	rpoB + katG
de Vos et al. [7]	Western cape (capetown)	2018	PTB	-	Flouro Type MTBDR and MTBDRplus	448/51,623 (0.9%)	244/43,580 (0.5%)	448/650 (69%)	-	13/2995 (0.4%)	35/3460 (1%)	163/2909 (5.6%)	rpoB: L511P (1), D516V (1), H526L (1), H526N (1) and L533P (1)	-	-	-	-
2. Rockwood et al. [8]	Western cape (Khayelitsha)	March 2013–2014	PTB	Prospective cohort study	GeneXpert MTB/RIF	306/51,623 (0.6%)	306/43,580 (1%)	-	-	17/2995 (0.5%)		-	rpoB: S531L (2) KatG: S315T (2)	-	13/2457 (0.5%)	-	-
3. Cox et al. [4]	Western cape (Khayelitsha)	2008–2017	Extra	Retrospective Cohort Study	Whole-genome sequencing (Illumina HiSeq 2500)	1274/51,623 (2.5%)	1274/43,580 (3%)	-	-	196/2995 (6.5%)	-	12/2909 (0.4%)	-	-	-	196/404 (48.5%)	-
4. Oostvogels et al. [5]	Western cape	2006–2017	PTB	Prospective study	MTBDRplus line probe assay Xpert MTB/RIF	748/51,623 (1.4%)	461/43,580 (1%)	-	461/3637 (12.7%)	461/2995 (15%)	461/3460 (12%)	197/2909 (7%)	-	-	-	-	-
5. Shah et al. [9]	Cape-town, Pune, Kampala, Lima	2017–2018	PTB	Prospective/multicenter study	Xpert MTB/RIF BD MAX MDR-TB assay	1053/5,1623 (1%)	984/43,580 (2.2%)	202/650 (31%)	232/3637 (6.4%)	27/2995 (1%)	10/3460 (0.3%)	-	D435Y (2), D435F (1), L430P (2), L452P (1)	16/885 (2%)	4/2457 (0.2%)	-	-
6. Brown et al. [10]	KZN (uMkhanyakude district, Ugu and Uthukela district)	2011–2014	PTB	Prospective study	Whole-genome sequencing	404/51,623 (1%)	318/43,580 (1%)	-	318/3637 (8.7%)	54/2995 (2%)		54/2909 (2%)	KatG: S315T (1)	-	-	-	-
7. Loiseau et al. [11]	Peru (Lima), Thailand (Bangkok), South Africa (Cape-Town), Kenya (Eldoret), Côte d’Ivoire (Abidjan), Botswana, DRC, Nigeria (Abuja), Tanzania (Bagamoyo)	-	PTB	Multicenter study	Whole-genome sequencing	312/51,623 (1%)	312/43,580 (1%)	-	312/3637 (8.6%)	276/2995 (9%)	282/3460 (8%)	246/2909 (8%)	rpoB S450L (282) KatG: S315T (250)	282/885 (32%)	276/2457 (11%)	-	-
8. Salaam-Dreyer et al. [12]	Cape town	2008–2017	PTB	Descriptive study	Xpert MTB/RIF	2161/51,623 (4%)	1119/43,580 (2.5%)	-	1119/3637 (31%)		2041/3460 (59%	899/2909 (31%)	rpoB L430P (32), rpoB S450L (73)	-	2041/2457 (83%)	-	-
9. Pitso et al. [13]	Free state (Fezile- Dabi, Lejweleputswa, Mangaung, Thabo Mofutsanyana, Xhariep)	2014–2016	PTB	Retrospective study	GenoType MTBDRplus	6648/51,623 (13%)	918/43,580 (2%)	-	918/3637 (25.2%)	123/2995 (4%)	587/3460 (17%)	918/2909 (31%)	-	587/885 (66%)	123/2457 (5%)	208/404 (51%)	-
10. Maningi et al. [14]	Western cape and Gauteng	1993–1995	-	-	Hain line probe assay (LPA) Illumina Miseq whole-genome sequencing (WGS) GenoType version 2 MTBDR*plus* assay (Hain Lifescience, Germany).	625/51,623 (1.2%)	-			5/2995 (0.2%)	44/3460 (1.3%)	171/2909 (6%)	rpoB (Thr480Ala (1), Gln253Arg (1), Val249Met (1), Val251Tyr (1), Val251Phe (2) KatG: (Trp477STOP (1), Gln88STOP (1), Trp198STOP (1), Trp412STOP (1)	-	-	-	-
11. Makhado et al. [15]	Gauteng, Mpumalanga, North west and Limpopo	2013–2016	-		Multiplex allele-specific PCR Deeplex-MycTB Whole-genome sequencing (Sanger sequencing)	37,644/51,623 (73%)	37,644/43,580 (86%)		277/3637 (7.6%)	1823/2995 (61%)	-	249/2909 (8%)	rpoB; IIe491phe (35) KatG: S315T (249)	-	-	-	-
Total	-	-	-	-	-	51,623/51,623 (100%)	43,580/51,623 (84%)	650/51,623 (1.2%)	3637/51,623 (7%)	2995/51,623 (6%)	3460/51,623 (7%)	2909/51,623 (6%)	rpoB: L511P (1), L533P (1), L430P (34)L452P (1), D516(1), D435Y (2), D435P (1), H526L (1), H526N (1), S531L (2), S450L (355), Thr480Ala (1), Gln 253 Arg (1), Val 249 Met (1), Val 251 Tyr (1), Val 251Phe (2), Ile 491 Phe (35) Total rpoB (441) KatG: S315T (502), Trp477STOP (1), Gln88STOP (1), Trp198STOP (1), Trp412STOP (1) Total KatG (506)	885/51,623 (2%)	2457/51,623 (5%)	404/51,623 (1%)	0

EXTRA = extrapulmonary tuberculosis; PTB = pulmonary tuberculosis; INH = isoniazid; RIF = Rifampicin; WT = wild-type; *rpoB, KatG, inHA*, *KatG* + *inhA, rpoB* + *katG* = gene mutation.

## Data Availability

Not applicable.

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
