# Peer review of "Isoniazid and Rifampicin Resistance-Conferring Mutations in Mycobacterium tuberculosis Isolates from South Africa"

_pathogens, 2023, doi:10.3390/pathogens12081015_

Round 1
Reviewer 1 Report
Title: Isoniazid and Rifampicin Resistance-Conferring Mutations in Mycobacterium tuberculosis Isolates from South Africa.
Authors: Traore et al.
MS: Pathogens-2525.
The authors have conducted a review of the literature on the most prevalent drug-resistance conferring mutations in M. tuberculosis strains in South Africa. The study included an analysis of 11 studies which met their inclusion criteria and focused on resistance to front-line treatment antibiotics rifampicin (RIF) and isoniazid (INH).
Main Points.
1. In the Summary and other places in the text (Page 11, line 32, Page 12, line 112) the authors mention S315T and Ser315Thr as if they were separate mutations but these surely are two ways of describing the same mutation in the KatG gene? S and T are the one-letter abbreviations for the amino acids serine and threonine respectively. I can see why this has come about as the authors have kept to the formats of the articles included. However, to a reader not in the field this might be construed as different mutations due to the way the findings are presented e.g.
“the predominant mutations were S315T (50%) and Ser315Thr (49%), leading to decreased activation of INH and compromising its efficacy in killing M. tb”.
2. P2 lines 52 -56.
“To further understand the overall proportion of phenotypic INH and RIF resistance explained by the existing single or canonical gene mutations, the estimated pooled prevalence of RIF resistance-associated gene mutations as well as the frequencies of co-occurring or multiple mutations have not been investigated”.
This sentence seems to be defining the aims of the present study, but ends up with the phrase “have not been investigated”.
Please clarify the intended meaning.
3. Table 1 is quite complicated and the text has wrapped in some columns. The effect of this is to make the important points in the table difficult to access with loss of impact. I suggest in any revision that the Table be split into two, with study details (numbered 1-11) in one and key data points relating to study number in the other.
4. Sections 3.2 and 3.3: There is a certain degree of repetition in these two sections, particularly between paragraphs beginning with line 137-144 and lines 147-153. These could be amalgamated and shortened without losing any detail.
5. The inclusion of one of the studies (Makhado NA, et al (2018) Outbreak of multidrug-resistant tuberculosis in South Africa undetected by WHO-endorsed commercial tests: an observational study. The Lancet Infectious Diseases; 18(12): 1350-1359)
might be seen as controversial as this study has received criticism in the peer-reviewed literature based on skewed sampling and various other methodological flaws (see Ismail et al, Lancet Infectious Diseases Dec 2018 https://doi.org/10.1016/S1473-3099(18)30715-1
Minor points and presentation.
Page 1, line 39: Authors, perhaps define InhA at first use.
Page 1, line 38 and 40: Like all bacterial genera and species, M. tuberculosis is usually italicized for easy recognition. It appears in the manuscript in several places in regular text (Summary, line 11; Introduction line 23; page 2, lines 73 and 82; Discussion page 10). The abbreviation could be used here as already defined earlier (line 23). In fact it is defined in 3 places Summary, Introduction and section 2.3. Consistency is needed, as both M. tb and M. tb appear in the manuscript.
Page 1, line 41. Maybe clarify that this is a global statistic.
P4. Line 120. Retrieved (past tense).
Page 4, Lines 129-130 no need for capital in “retrospective”.
Page 4, line 144. While or whilst none etc?
P11, line 41. The reference for the Indian study mentioned should be given here.
Page 11, line 72. An inhA promoter region ?
The quality of the English language and grammar is generally good. There are a few minor corrections needed and specific suggestions have been mentioned in the review.
Reviewer 2 Report
The manuscript entitled “Isoniazid and Rifampicin Resistance-Conferring Mutations in Mycobacterium tuberculosis Isolates from South Africa” deals with the prevalence of gene mutations associated with RIF and INH resistance, which shows a high prevalence of some specific mutations. The authors have revealed link of these findings to drug resistance against and have contributed to the understanding of drug-resistant TB. The manuscript is well-written, and the discussion are supported by data. Also, literature citation is adequate and figures and tables are mostly suitable for the readers. Howerver, some points should be addressed.
Major points
Point 1:
Figures and tables: They can be smaller. Most of them are relatively big and over the size.
Point 2:
Additional conclusion can be more carefully written to emphasize the valuable results.
Minor points
'Mycobacterium tuberculosis' is better written in italic font.
Reviewer 3 Report
The manuscript presents the systematic review of mapping of muations those results in resistance to frontline TB drugs Rifampicin and Isoniazid in clincal isolated from South African patients. Although, authors have down a significant work to observe and analyze the previous published studies and do a systematic review there are still much scope to improve. Following are some if my suggestions:
1) Metaanalysis is a most important part of systamic reviews as they give you a combined birds eye view of previous studies together and a chance to compare those. Therefore, more efforts has to go on explaining the data obatined from this analysis which here is Forest plot. Authors should elaborate more on the interpretation of Forest plot obatined from the data.
2) Authors should provide an possible explaination for the observed heterogeneity observed between the studies. Also, if the heterogeneity is not by chance event is a real observation than should the relation between the mutations and drug resistance phenotype be believed? please clarify.
3) In the Forest plot, for most of the studies the line of null effect is between the 95% confidence interval which means there is the chance that there is no relation between the mutation and the drug resistance phenotype. Please comment.
4) In line 139, it is mentioned that only 650 pateints have gone through DST but in further part on the manuscript mutation profile of 3637 pateints are drug resistant is being mentioned. This is confusing to me, please clarify.
Round 2
Reviewer 1 Report
Dear Authors,
Thank you for revising your manuscript along suggested lines. This reviewer has read through the draft and now has only minor revisions to improve the paper.
1. Introduction, line 39.
inhA is enoyl-ACP reductase, a component of the mycobacterial FAS-II system of mycolic acid synthesis. It is inhibited by isoniazid when this prodrug is itself activated by catalase-peroxidase (KatG).
The inhibin subunit Alpha mentioned here is not involved in mycolic acid synthesis in mycobacteria but in reproductive gynaecology.
2. I notice on reading the MS again, that XDR-TB appears in the Conclusions, not having been mentioned previously. Strictly speaking the definitions of MDR and XDR differ.
MDR is defined as acquired non-susceptibility to at least one agent in three or more antimicrobial categories, XDR is defined as non-susceptibility to at least one agent in all but two or fewer antimicrobial categorie. Authors should decide if XDR is needed here and/or earlier in the MS and defined upon first use.
3. Reference 26 - Seifert has spurious comma.
Reviewer 3 Report
The authors have addressed all my comments and have made the required changes to the original manuscript. I congratulate the team for the good work.
Author Response
Thank you very much. you've put in a lot of time and effort into building this review paper. we greatly appreciate your thoughts and recommendations for improving the work. we are truly grateful!!